# Percutaneous Image-Guided Vertebral Fixation in Cancer-Related Vertebral Compression Fractures: A Case Series Study

**DOI:** 10.3390/medicina57090907

**Published:** 2021-08-30

**Authors:** Francois H. Cornelis, Leo Razakamanantsoa, Mohamed Ben Ammar, Milan Najdawi, Francois Gardavaud, Sanaa El-Mouhadi, Matthias Barral

**Affiliations:** 1Department of Interventional Radiology and Oncology, Tenon Hospital, Sorbonne Université, 75020 Paris, France; leo.razakamanantsoa@aphp.fr (L.R.); mohamed.benammar@aphp.fr (M.B.A.); milan.najdawi@aphp.fr (M.N.); francois.gardavaud@aphp.fr (F.G.); matthias.barral@aphp.fr (M.B.); 2Department of Radiology, Saint Antoine Hospital, Sorbonne Université, 75012 Paris, France; sanaa.el-mouhadi@aphp.fr

**Keywords:** back pain, kyphosis, spinal fractures, vertebroplasty, cancer

## Abstract

*Background and objectives:* Cancer-related vertebral compression fractures (VCF) may cause debilitating back pain and instability, affecting the quality of life of cancer patients. To further drive cement deposition during vertebroplasty, the aims of this restrospective case series study were to report the feasibility, safety and short term efficacy (≤6 months) of percutaneous vertebral fixation in cancer-related vertebral compression fractures using various intravertebral implants. *Methods:* All consecutive cancer patients treated with percutaneous vertebral fixation for VCF were retrospectively included. Various devices were inserted percutaneously under image guidance and filled by cement. Descriptive statistics were used and a matched paired analysis of pain scores was performed to assess for changes following interventions. *Results:* A total of 18 consecutive patients (12 women (66.6%) and 6 men (33.3%); mean age 59.7 ± 15.5 years) were included. A total of 42 devices were inserted in 8 thoracic and 16 lumbar vertebrae. Visual analogue scale measurement significantly improved from 5.6 ± 1.8 preoperatively to 1.5 ± 1.7 at 1 week (*p* < 0.01) and to 1.5 ± 1.3 at 6 months (*p* < 0.01). No severe adverse events were observed, but three adjacent fractures occurred between 1 week and 5 months after implantation. *Conclusions:* Percutaneous vertebral fixation of cancer-related VCF is feasible and safe and allows pain relief.

## 1. Introduction

Debilitating back pain and instability may be observed in cases of cancer-related vertebral compression fractures (VCF), affecting the quality of life of cancer patients [1]. It may lead to reduced overall survival [2]. Despite concerns about their safety and efficacy, interventional radiology procedures such as percutaneous vertebroplasty improve the clinical outcomes of cancer patients [3]. However, vertebroplasty alone appears insufficient to adequately drive the cement deposition and to correct the structural deformities observed after VCF [4]. Cement leakage may be observed in cases of lytic lesions involving the posterior wall of the vertebral body, and the height of the vertebrae is often not restored, leading to further instability [5].

So far, several other percutaneous techniques have been proposed to further improve cement deposition and stabilization of cancer-related VCFs, mainly by using intravertebral devices such as balloon kyphoplasty, and more recently, implants [6,7,8,9]. Such implants have been already evaluated in osteoporotic VCFs but scarcely in cancer patients [10]. The purpose of this case-series study was to report the feasibility, safety and short-term efficacy (≤6 months) of percutaneous vertebral fixation using various intervertebral implants in cancer-related VCFs.

## 2. Materials and Methods

From 2019 to 2021, all consecutive cancer patients with symptomatic VCFs who had vertebral fixation using intravertebral implants were retrospectively included in this monocentric institutional review board-approved case series study. Patients were considered eligible for inclusion if they presented with VCF involving the lower thoracic and/or lumbar vertebrae; collapse of vertebrae <50%; spinal instability neoplastic score (SINS) ≥7 [11,12]; and had at least a 6-month clinical and radiological follow-up.

Patients were excluded if vertebral body collapse was >50%; if the location was on the cervical spine or rigid spine (S2-5); if patients presented with neurological symptoms requiring urgent open surgical decompression; or if patients had a general contraindication to anesthesia. All patients gave their written informed consent before treatment. Population characteristics are summarized in Table 1.

Preoperative evaluation included clinical examination and an imaging exploration including a Computed Tomography (CT) scan and Magnetic Resonance Imaging (MRI) when feasible.

All interventions were performed under general anesthesia and cone-beam computed tomography guidance (Innova IGS540, General Electrics, Milwaukee, WI, USA). One or two 10 G bone trocarts (Stryker, Kalamazoo, MI, USA) were directly inserted into the vertebral body via a mono or a bilateral transpedicular approach in the lumbar spine or an intercostal approach in the thoracic spine. A blunt guidewire was placed, and a device-specific drill mounted on a working cannula was advanced manually on it into the vertebral body until the desired position was reached, located 5 mm from the anterior wall as assessed on a lateral X-rays view. After the drill and the guidewire were removed, the devices were inserted through the cannula. Three different intravertebral implants were used: Spinejack^®^ (Stryker, Kalamazoo, MI, USA) (Figure 1); V-Strut^®^ (Hyprevention, Pessac, France) (Figure 2); and KIVA^®^ (IZI medical, Owing Mills, MD, USA) (Figure 3). These devices are detailed in [6] and their differences in [4]. They were carefully deployed in order to ensure correct placement and control potential posterior wall protrusion. The implants were released and filled by Poly-Methyl-Methacrylate (PMMA) bone cement (Vertaplex HV, Stryker, Kalamazoo, MI, USA for Spinejack^®^ or Osteofix, IZI medical, Owing Mills, MD, USA for KIVA^®^ and V-strut^®^), slowly injected through a cannula.

Procedure-related adverse events were systematically assessed according to Cardiovascular and Interventional Radiological Society of Europe (CIRSE) guidelines [13]. A clinical follow-up was performed after the intervention to evaluate pain relief and functional improvements, in addition to an imaging exploration.

## 3. Results

A total of 18 patients were included in this study (12 women [66.6%] and 6 men (33.3%), mean age: 59.7 ± 15.5 years (interquartile range (IQR): 53.3–70.8). A total of 7 patients received radiation therapy before treatment and 5 after.

A total of 42 implants (30 Spinejack^®^, 10 V-strut^®^ and 2 Kiva^®^) were inserted in 24 levels: 16 lumbar (66.7%) and 8 thoracic vertebrae (33.3%). Synchronous fractures were treated by vertebroplasty during the same procedure in 9 patients (50%) for a total of 19 levels in upper vertebra in 6 patients (33.3%) and lower vertebra in 12 patients (66.6%). One patient received electrochemotherapy of the vertebral tumor at the same time.

Mean fluoroscopy time (FT) was 15.6 ± 7.9 min (11–19.8), mean kerma area product (KAP) was 34.6 ± 23.1 Gy.cm^2^ (20.2–42.5) and mean air kerma dose (AK) was 358.3 ± 240.6 mGy (218.5–422.5).

No patients were lost to the follow-up at 6 months. Mean visual analogue scale (VAS) score significantly improved from 5.6 ± 1.8 preoperatively to 1.5 ± 1.7 at 1 week (*p* < 0.01) and to 1.5 ± 1.3 at 6 months (*p* < 0.01) (Figure 4). At the follow-up CT scan, seven patients (38.9%) presented PMMA cement venous leakages without any clinical consequence (ten posterolateral, three posterior, one anterior, two intradiscal leakages) (grade 1 complications). Vertebral height restoration was observed only after Spinejack^®^ implantation in eight patients (44.4% total but 72.7% of the patients who received Spinejack^®^), with a gain in height of 5.7 ± 2.2 mm in mean (4.5–7).

No major procedure-related complications (grade ≥3) occurred in the immediate post-operative period. Three patients (16.7%) developed a secondary adjacent level fracture: at one week of bipedicular Spinejack^®^ implantation (Patient 9, grade 3 complication), at 1 month (Patient 5, grade 2 complication) and at 5 months (Patient 3, grade 2 complication) of bipedicular V-strut^®^ implantations. Two additional procedures were performed in two of these patients: a vertebroplasty in Patient 3 and a bipedicular Spinejack^®^ insertion in Patient 9.

## 4. Discussion

In cancer-related VCFs, vertebral fixation appears to be feasible, safe and effective to achieve short-term pain palliation, whatever the intravertebral implants used. By improving cement deposition and stability, intravertebral implantation of these devices avoids further complications, even when vertebrae are involved extensively by the tumors.

Performing vertebral fixation appears not challenging and may be further implemented in routine. However, these procedures required general anesthesia [14,15]. The mean KAP 34.6 ± 23.1 Gy.cm^2^ (20.2–42.5), mean AK 358.3 ± 240.6 mGy (218.5–422.5) and mean FT of 15.6 ± 7.9 min (11–19.8) calculated in this study were lower to the reference strandards required by the European Directive 2013/59/Euratom in interventional radiology for a vertebroplasty in 1 level (KAP: 60 Gy.cm^2^, AK: 610 mGy and FT: 9 min) or 3 or more levels (110 Gy.cm^2^, 1160 mGy and 14 min) [16,17]. While further studies are needed to validate these results, the use of such intravertebral implants may further help to standardize the procedures without increasing dosage [18].

In terms of adverse events, few asymptomatic cement leakages were observed. In the case of V-strut^®^ implantation, it was probably related to the low viscosity of the cement used during the procedure. A careful selection and preparation of cement in addition to an improved imaging monitoring may help to reduce such leakages. Besides, three patients had adjacent fractures after device implantation. Such fractures are well-known complications that have already been reported in the literature after vertebroplasty or vertebral augmentation [19]. These are possibly related to the progression of the disease but also to the modifications of the local constraints induced by the cement and the intravertebral devices. Further studies on larger cohorts are needed to confirm the cause of such events, which might help to better select the patients for vertebral fixation [20,21].

Several limitations of this study may be considered. While the purpose of all the devices remained similar, all three devices used in this study have different designs, which may explain some of the differences observed. KIVA^®^ ensures structural correction by filling large intravertebral lesions [10]. V-strut^®^ aims to transfer the load to the posterior column, facilitating stabilization [8,22]. Spinejack^®^ is designed to homogeneously restore the vertebral height and correct the kyphosis by creating a unidirectional craniocaudal thrust force focused on the points of compression without unnecessary damages to the trabecular bone [19,23]. Despite that pain relief was obtained in most of the patients, it remains difficult to draw definitive conclusions about the clinical efficacy on pain for the technique of fixation by itself, especially in such a small population, as patients with cancer-related VCFs usually have various treatments during the course of the disease such as morphinics, ablation or radiation therapy [2]. However, a significant effect of vertebral augmentation for cancer-related VCFs was demonstrated in a review compared to non-surgical management, radiofrequency ablation or chemotherapy alone [10]. Other limitations were related to the retrospective and monocentric design of the case series as well as the short-term follow-up.

## 5. Conclusions

Percutaneous vertebral fixation of cancer-related VCFs is feasible and safe, whatever the intravertebral implants used. Fixation allows short-term pain relief. Further larger series are needed to better assess the performance of these procedures and select the adequate technique for a specific patient.

## Figures and Tables

**Figure 1 medicina-57-00907-f001:**
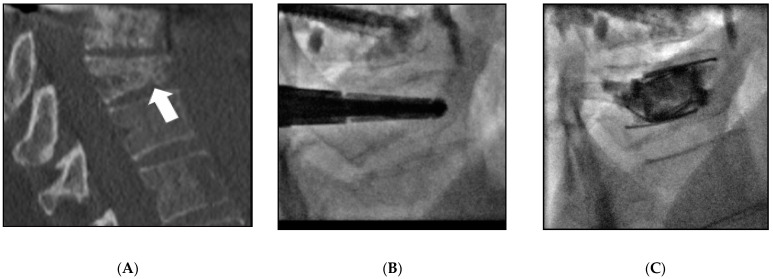
Spinejack^®^ implantation in a 49-year-old woman (Patient 18) with breast carcinoma metastasis of L2. (**A**) Sagittal reconstruction of the computed tomography, arrow. (**B**) The two devices were inserted under fluoroscopy into the vertebral body via a bilateral transpedicular approach. (**C**) After deployment and cement injection, the height of the vertebrae was restored.

**Figure 2 medicina-57-00907-f002:**
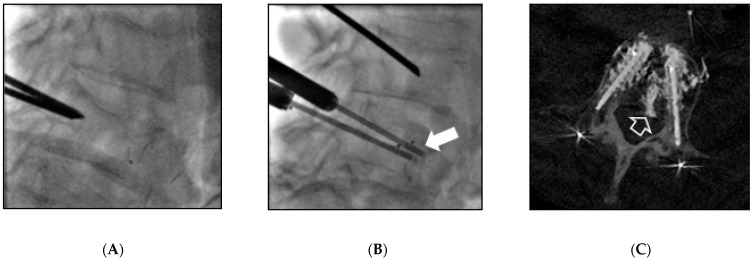
V-strut^®^ implantation in a 54-year-old woman (Patient 5) with breast carcinoma metastasis of L4. (**A**) The two devices were inserted into the vertebral body via a bilateral transpedicular approach until the desired position of 5 mm from the anterior wall, arrow. (**B**) A posterior cement leakage was observed on final cone beam computed tomography reconstruction, arrow. (**C**) Results are presented as means ± standard deviations. A matched paired analysis of visual analogue scale scores was performed to assess for changes in pain following interventions using Student’s t-test. Results were considered statistically significant when *p*-values < 0.05. Data analysis was performed using Stata 20.0 (StataCorp LLC, College Drive, TX, USA).

**Figure 3 medicina-57-00907-f003:**
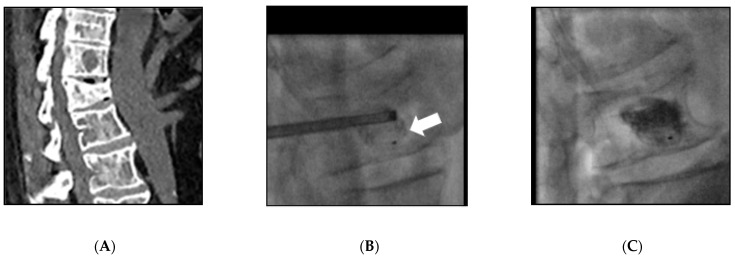
KIVA^®^ implantation in L2 in a 57-year-old man (Patient 16) with myeloma. (**A**) sagittal reconstruction of CT scan. (**B**) The device (arrow) was inserted into the vertebral body under fluoroscopy via a monolateral transpedicular approach. (**C**) After deployment of the device, the cement was injected via injection.

**Figure 4 medicina-57-00907-f004:**
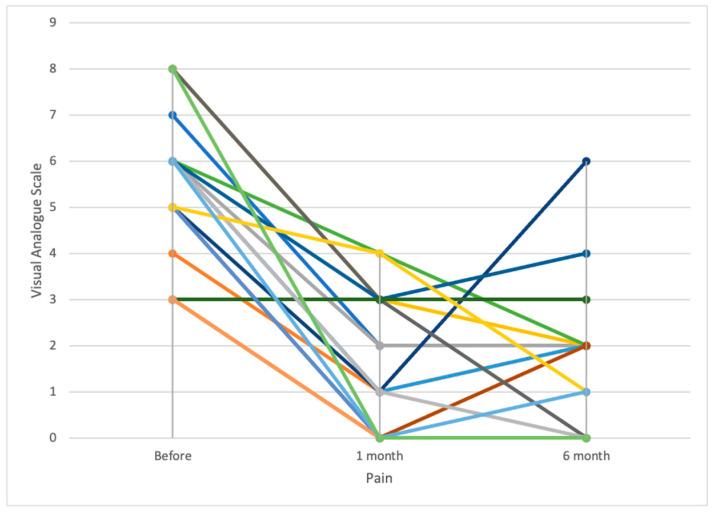
Pain evolution at 1 and 6 months after treatment, assessed by visual analogue scale scores for each patient. Mean visual analogue scale score significantly improved from 5.6 ± 1.8 preoperatively to 1.5 ± 1.7 at 1 week (*p* < 0.01) and to 1.5 ± 1.3 at 6 months (*p* < 0.01).

**Table 1 medicina-57-00907-t001:** Population characteristics and procedure outcomes.

Patients	Age	Gender	Location of Primary Tumor	Location	SINS	Device (Number of Device)	Associated Procedures	Location	Adverse Events of Vertebral Fixation (Grade)	Treatment of Adverse Event (Number of Device)
1	59	Female	Breast	L2	10	V-strut^®^ (2)	Electrochemotherapy	L2	Cement leakage (1)	-
2	53	Female	Breast	L4	8	V-strut^®^ (2)	-	-	-	-
3	64	Male	Lung	L2	9	V-strut^®^ (2)	Vertebroplasty	L3	VCF L4&L5 (2) Cement leakage (1)	Vertebroplasty
4	58	Male	Lung	L4	9	V-strut^®^ (2)	-	-	Cement leakage (1)	-
5	54	Female	Breast	L4	10	V-strut^®^ (2)	Vertebroplasty	L3	VCF L5 (2) Cement leakage (1)	-
6	48	Male	Lung	L1&L4	9	SpineJack^®^ (4)	Vertebroplasty	T9 T10 T11 T12	-	-
7	62	Male	Lung	L3	9	SpineJack^®^ (2)	Vertebroplasty	T2 T3 T4	-	-
8	40	Female	Breast	T8	7	SpineJack^®^ (2)	-	-	-	-
9	73	Female	Breast	L1	10	SpineJack^®^ (2)	-	-	VCF L2 (3) Cement leakage (1)	Spinejack^®^ (2)
10	56	Female	Ovarian	T12	12	SpineJack^®^ (2)	Vertebroplasty	T10 T11 L1	-	-
11	79	Female	Lung	L1&L2	01	SpineJack^®^ (4)	-	-	-	-
12	23	Female	Osteosarcoma	T12&L1	9	SpineJack^®^ (2)	Vertebroplasty	C7 T3 T5	Cement leakage (1)	-
13	89	Female	Ovarian	T11&L1	10	SpineJack^®^ (2)	Vertebroplasty	T12	-	-
14	74	Female	Ovarian	T12&L1	10	SpineJack^®^ (4)	Vertebroplasty	L2	Cement leakage (1)	-
15	59	Female	Breast	T9	9	Kiva^®^ (1)	-	-	-	-
16	57	Male	Myeloma	L2	7	Kiva^®^ (1)	-	-	-	-
17	78	Male	Lung	T12	7	SpineJack^®^ (2)	Vertebroplasty	T3 T4	-	-
18	49	Female	Breast	T10	7	Spinejack^®^ (2)	-	-	-	-

Note—L: lumbar; SINS: spinal instability neoplastic score; T: thoracic; C: cervical; VCF: vertebral compression fracture.

## Data Availability

The data of the study are present at the Departement of Interventional Radiology and Oncology at Tenon Hospital. The data will be available upon reasonable request.

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
