# Peer review of "Percutaneous Image-Guided Vertebral Fixation in Cancer-Related Vertebral Compression Fractures: A Case Series Study"

_medicina, 2021, doi:10.3390/medicina57090907_

Round 1

Reviewer 1 Report

Title of Manuscript: Percutaneous image-guided vertebral fixation in cancer-related vertebral compression fractures: an evaluation of 6-month out-comes

Manuscript No. medicina-1279206

Comment: Authors showed that vertebral fixation of cancer-related VCFs is feasible and safe. And the efficacy was shown in VAS score.

  1. If authors would like to show the feasible and safe way when conducting vertebroplasty for metastatic spinal tumor, please make control group as osteoporotic vertebral fracture and so on.
  2. Authors showed grade 1 complications. Please state the reasons clearly. It had the possibility not only for the viscosity but also for the lytic lesion.
  3. Authors should describe the difference of devices in methods section and how to select the devices for each patient.
  4. Authors showed the correction of the vertebral body height and the pain score using VAS. Please describe the relationship between the change of vertebral body height and pain scores.
  5. Authors should show the permission from the author described SINS.
  6. Authors showed insufficient abstract as follows;

Abstract section, P1 line 3: Various de-vices were inserted using a similar percutaneous image-guided procedure.

This sentence did not show at all what you did. Authors did vertebroplasty using 3 devices. Authors should state more clearly what devices you inserted.

Abstract section, P1 line 4: Descriptive statistics were used.

This sentence did not show at all what you did. Authors should state more clearly what kind of statistics authors used.

Author Response

Authors: The authors would like to thank both the reviewers and the editor for their positive evaluation of the manuscript. We have revised the manuscript as per the Reviewers’ recommendations. We respond point-by-point to the comments below. 1. As it case series study, it has to be noted somewhere, preferably in the title. Authors: the title has been changed accordingly. 2. The first heading in the abstract has to be "Background and objectives". Please include a brief background. Authors: the abstract has been edited accordingly. 3. The information on statistical analysis has to be clarified. The authors state that "Results are presented as means ± standard deviations and analyzed using Wilcoxon test, depending on the data distribution". First, it seems that the data are expressed as median with IQR. Second, as the authors expressed the data differently, this sentence has to be changed and updated. Have the authors checked how their data are distributed? They do not mention this. Based on the normality of data distribution, the data have to be expressed differently. When data are non-normally distributed, the median and range (or interquartile range) provide a better summary of the distribution, when normally, mean with SD. Based on data distribution, appropriate tests have to be applied (parametric or non-parametric). Please clarify this. Authors: the statistical analysis has been corrected as suggested. 4. According to the iThenticate report, there are sentences taken from https://link.springer.com/article/10.1007%2Fs00330-018-5544-6 and used in this manuscript as the authors' thoughts. Please rephrase these sentences." Authors: the manuscript has been edited. This article was already cited as a reference in the initial submission.

Reviewer 1: Comment: Authors showed that vertebral fixation of cancer-related VCFs is feasible and safe. And the efficacy was shown in VAS score. 1. If authors would like to show the feasible and safe way when conducting vertebroplasty for metastatic spinal tumor, please make control group as osteoporotic vertebral fracture and so on. Authors: Thank you for this comment. This study is a descriptive case series on cancer patients evaluating the feasibility and the safety of fixation in these specific patients. Therefore, a control group was not used for comparison. Further study should compare the efficacy of fixation to vertebroplasty alone in cancer patients instead of porotic patients as outcomes differ dramatically. 2. Authors showed grade 1 complications. Please state the reasons clearly. It had the possibility not only for the viscosity but also for the lytic lesion. Authors: Complications were reported following CIRSE’s recommendations which grades complications from 1 to 5. The related reference was mentioned in materials and methods. 3. Authors should describe the difference of devices in methods section and how to select the devices for each patient. Authors: All patients presented similar inclusion and exclusion criteria. The selection was only based on the availability of the devices at the time of implantation. All devices have the same purpose and were extensively detailed in REF 4: Cornelis FH, Joly Q, Nouri-Neuville M, et al (2019) Innovative spine implants for improved augmentation and stability in neoplastic vertebral compression fracture. Med. doi: 10.3390/medicina55080426. A sentence has been added for clarification. 4. Authors showed the correction of the vertebral body height and the pain score using VAS. Please describe the relationship between the change of vertebral body height and pain scores. Authors: Such evaluation would be evaluated in a future study as concerning only a small subset of the patients included, and only patients treated with SpineJack. The relationship between change of vertebral body height and pain score was not evaluated therefore in this descriptive case series. 5. Authors should show the permission from the author described SINS. Authors: SINS is a routinely used classification system, already cited in many publications. We cited the authors. We added the table 1 to improve the readership but we can remove it if needed. 6. Authors showed insufficient abstract as follows; Abstract section, P1 line 3: Various devices were inserted using a similar percutaneous image-guided procedure. This sentence did not show at all what you did. Authors did vertebroplasty using 3 devices. Authors should state more clearly what devices you inserted. Authors: we edited the text according to this comment. Abstract section, P1 line 4: Descriptive statistics were used. This sentence did not show at all what you did. Authors should state more clearly what kind of statistics authors used. Authors: we edited the text according to this comment.

Reviewer 2 Report

The authors present a retrospective, single institution analysis of the feasibility, safety, and short term efficacy of vertebral fixation in cancer-related vertebral compression fractures. The study is of interest to oncologists managing malignancies affecting the spine.

Minor comments:

1) Did the authors utilized a matched paired analysis of VAS scores to assess for changes following interventions?

2) Did patients receive radiation therapy to the spine either before or after the interventions? If so, can the authors elaborate on whether pain improvement was related to receipt of radiotherapy?

Author Response

Authors: The authors would like to thank both the reviewers and the editor for their positive evaluation of the manuscript. We have revised the manuscript as per the Reviewers’ recommendations. We respond point-by-point to the comments below. 1. As it case series study, it has to be noted somewhere, preferably in the title. Authors: the title has been changed accordingly. 2. The first heading in the abstract has to be "Background and objectives". Please include a brief background. Authors: the abstract has been edited accordingly. 3. The information on statistical analysis has to be clarified. The authors state that "Results are presented as means ± standard deviations and analyzed using Wilcoxon test, depending on the data distribution". First, it seems that the data are expressed as median with IQR. Second, as the authors expressed the data differently, this sentence has to be changed and updated. Have the authors checked how their data are distributed? They do not mention this. Based on the normality of data distribution, the data have to be expressed differently. When data are non-normally distributed, the median and range (or interquartile range) provide a better summary of the distribution, when normally, mean with SD. Based on data distribution, appropriate tests have to be applied (parametric or non-parametric). Please clarify this. Authors: the statistical analysis has been corrected as suggested. 4. According to the iThenticate report, there are sentences taken from https://link.springer.com/article/10.1007%2Fs00330-018-5544-6 and used in this manuscript as the authors' thoughts. Please rephrase these sentences." Authors: the manuscript has been edited. This article was already cited as a reference in the initial submission.

Reviewer 2 The authors present a retrospective, single institution analysis of the feasibility, safety, and short term efficacy of vertebral fixation in cancer-related vertebral compression fractures. The study is of interest to oncologists managing malignancies affecting the spine. Minor comments: 1) Did the authors utilized a matched paired analysis of VAS scores to assess for changes following interventions? Authors: we performed a matched paired analysis. We edited the manuscript accordingly. 2) Did patients receive radiation therapy to the spine either before or after the interventions? If so, can the authors elaborate on whether pain improvement was related to receipt of radiotherapy? Authors: most of the patients received radiation therapy: 7 before treatment and 5 after. It was not possible to address the question rose by the reviewer. We edited the manuscript to mention this limit.

Round 2

Reviewer 1 Report

Title of Manuscript: Percutaneous image-guided vertebral fixation in cancer-related vertebral compression fractures: a case series study

Manuscript No. medicina-1279206

  1. Authors should withdraw Table 1. This table is not authors’ original.
  2. Authors should show the patient’s background status when authors conducted matched pair analysis. At least, the number conducting matched pair analysis should show.

Author Response

Dear Editor, please find the answers to the comments below.

Authors should withdraw Table 1. This table is not authors’ original.

Authors : we deleted table 1 as requested.

Authors should show the patient’s background status when authors conducted matched pair analysis. At least, the number conducting matched pair analysis should show.

Authors : we added figure 4 to provide concisely the data supporting the analysis without decreasing the readership of the table. Please let us know if this option is acceptable.

Reviewer 2 Report

The authors have appropriately addressed my concerns. 

Author Response

Thank you.